# A Systematic Review and Meta-Analysis of Nature-Based Mindfulness: Effects of Moving Mindfulness Training into an Outdoor Natural Setting

**DOI:** 10.3390/ijerph16173202

**Published:** 2019-09-02

**Authors:** Dorthe Djernis, Inger Lerstrup, Dorthe Poulsen, Ulrika Stigsdotter, Jesper Dahlgaard, Mia O’Toole

**Affiliations:** 1Department of Geoscience and Natural Resource Management, University of Copenhagen, C 1958 Frederiksberg, Denmark; 2Department of Landscape Architecture and Management, Swedish University of Agricultural Sciences, 230 53 Alnarp, Sweden; 3Department of Clinical Medicine, Aarhus University, N 8200 Aarhus, Denmark; 4Research Center for Health and Welfare Technology, VIA University College, N 8200 Aarhus, Denmark; 5Department of Psychology and Behavioural Sciences, Aarhus University, C 8000 Aarhus, Denmark

**Keywords:** nature contact, mindfulness, meditation, nature-based therapy, intervention

## Abstract

Research has proven that both mindfulness training and exposure to nature have positive health effects. The purpose of this study was to systematically review quantitative studies of mindfulness interventions conducted in nature (nature-based mindfulness), and to analyze the effects through meta-analyses. Electronic searches revealed a total of 25 studies to be included, examining 2990 participants. Three analyses were conducted: Nature-based mindfulness interventions evaluated as open trials (k = 13), nature-based mindfulness compared with groups in non-active control conditions (k = 5), and nature-based mindfulness compared with similar interventions but without contact with nature (k = 7). The overall combined psychological, physiological, and interpersonal effects from pre- to post-intervention were statistically significant and of medium size (*g* = 0.54, *p* < 0.001). Moderation analyses showed that natural environments characterized as forests/wild nature obtained larger numerical effects than environments characterized as gardens/parks, as did informal mindfulness compared with formal mindfulness. The small number of studies included, as well as the heterogeneity and generally low quality of the studies, must be taken into consideration when the results are interpreted. PROSPERO registration number: CRD42017065639.

## 1. Introduction

Recent research has paid increasing attention to the healing effects of both mindfulness training and exposure to nature, and research at the intersection between environmental psychology and meditation science seems to be a growing field [1], as does the variety of applications available [2]. Possible cumulative or synergistic effects may exist. An example is the correlation between the feeling of being connected to nature and the capacity to be mindful (i.e., trait mindfulness), which has been documented by Schutte and Malouff [3]. Another example is the theory-based hypothesis that meditation training and exposure to nature complement each other when combined [4], and are not merely the addition of independent effects.

### 1.1. Mindfulness

Mindfulness is considered to involve the intentional regulation of attention with an attitude of non-judgment, openness, curiosity, and acceptance of one’s current experience [5]. While state mindfulness may change rapidly, trait mindfulness can be enhanced through meditation. In programs such as mindfulness-based stress reduction (MBSR) and mindfulness-based cognitive therapy (MBCT), most formal meditations can be characterized as “attention training”: Attentional regulation is cultivated by intentionally bringing the attention back to a chosen object (concentration meditation), by paying attention to whatever thoughts or sensations unfold by themselves in the mind (open-awareness meditation), or by being aware of the awareness itself [6,7]. Informal mindfulness is also part of the MBSR curriculum, and is cultivated while one engages in everyday activities [6]. Short mindfulness interventions have been documented to enhance state mindfulness [8] but most research on mindfulness training concerns the effects of programs such as the eight-week MBSR and MBCT. These have been proven to affect health positively with regard to psychological [9,10], physical [11,12], and social functioning [13,14]. With regard to the mechanisms that underlie mindfulness interventions comparable to MBSR, evidence has been found regarding cognitive and emotional reactivity, mindfulness, and ruminative negative thinking, and preliminary but insufficient evidence has been found regarding self-compassion and psychological flexibility [15].

### 1.2. Natural Environments and Exposure to Nature

Nature can be defined as “areas containing elements of living systems that include plants and nonhuman animals across a range of scales and degrees of human management, from a small urban park through to relatively ‘pristine wilderness’” [16] (p. 120). For the purposes of data extraction, this article uses Mausner’s [17] categories of natural environments, which seem appropriate for the task and compatible with Bratman et al.’s definition.

It has been shown that spending time in nature promotes health, prevents health problems such as stress, depression, and anxiety [18,19,20], and improves immune functioning [21] and interpersonal functioning [22]. The pathways to positive health effects may be via physical and social activity as well as improved air quality [23]. Improved immune functioning from contact with microbial or other antigens [21] or exposure to certain natural substances such as phytoncides from trees [24] may also function as a mechanism for associated health benefits. The most widely investigated psychological pathways are attention restoration and stress reduction, but other factors, such as emotion regulation and feelings of awe or mystery, may mediate positive health outcomes [20]. A number of nature-based interventions have been developed that draw on the healing effects believed to be inherent to nature, including forest therapy, e.g., [25,26], wilderness therapy e.g., [27], adventure therapy e.g., [28], and horticultural therapy e.g., [29].

### 1.3. Possible Interactions between Mindfulness and Exposure to Nature

Attention is a psychological mechanism suggested to underlie the positive effects of both mindfulness training as a component of trait mindfulness [30], and exposure to nature [31] as part of attention restoration. Kaplan’s attention restoration theory (ART) is one of the most widely applied and empirically supported theories about the benefits of exposure to nature for mental health [31,32,33]. According to ART, directed attention is a limited resource, but it may nonetheless be restored. ART suggests that exposure to nature can support the restorative process [34,35], in part because such natural settings are often physically distant from one’s stressful everyday life, and also because nature promotes so-called soft fascination, which is defined as effortless attention drawn to fascinating objects. Although several environments and settings might potentially foster soft fascination, it has been suggested that natural environments are particularly well suited, as they inherently possess patterns that are sufficiently extended, rich, and coherent to engage the mind, which is believed to enable fascination in an “undramatic fashion” [35] (p. 174). ART is most often applied in relation to human-nature contact, but Kaplan [4] argues that similar mental processes may also occur during mindfulness meditation. The meditator is often distanced physically or mentally from everyday life, and can become softly fascinated as he or she effortlessly observes the stream of sensations, feelings, and thoughts. In order to be restorative, Kaplan argues that both exposure to nature and meditation need to fit the individual’s inclinations, motivations, and capabilities [4]. He suggests in particular that the untrained meditator, who tends to use effort in meditation, will have easier access to soft fascination in a restorative environment [4]. This may be true whether the meditation is part of a manual-based mindfulness program or a single guided meditation.

Another possible interaction between attention restoration and mindfulness is that paying attention to the environment is necessary for soft fascination to occur. This was shown in a study by Jiang, Schmillen, and Sullivan [36]: Students in a natural environment who were occupied with their portable electronic devices did not restore their attention in the same way as students in the same natural environment with no such devices. However, even when one is willing to pay attention to nature, the mind can easily wander, and the recollection of present-moment awareness may be challenging. This capacity is cultivated through mindfulness, which would therefore seem to support the acquisition of the potential health benefits of exposure to nature.

As research including both mindfulness and nature is a growing field, and common mechanisms and interactions seem to exist, it may be timely to make an initial synthesis in order to assess the quality and extent of research in the field. To our knowledge, no systematic review or meta-analysis currently exists that investigates the potential effects of nature-based mindfulness (mindfulness conducted in nature).

### 1.4. Aims

The primary aim of this study was to systematically review all existing studies investigating nature-based mindfulness interventions, and to quantify the results through meta-analyses. The hypothesis was that, due to the beneficial health-promoting effects of mindfulness and exposure to nature, combinations of mindfulness and nature evaluated in open trials would be positive, and the effects of nature-based mindfulness would be superior to passive control conditions, to mindfulness in non-natural settings, and to interventions in natural settings without mindfulness. Specifically, manual-based stand-alone mindfulness conducted in nature was to be evaluated, with the hypothesis that exposure to nature would positively affect the outcomes. To qualify the results, the additional aims were to explore the potential moderating effect of 1) the type of natural setting and 2) the type of mindfulness practice.

## 2. Materials and Methods

The study was preregistered in PROSPERO (registration number CRD42017065639).

### 2.1. Selection Criteria

The PICOS approach [37] was used to evaluate studies’ eligibility.

#### 2.1.1. Population

Adults and adolescents (>12 years) with or without a formal mental or physical diagnosis were included.

#### 2.1.2. Intervention

Interventions included both exposure to nature as defined by Mausner [17] and guided mindfulness, defined as “paying attention on purpose, in the present moment.” We left out one criterion often employed in the definition of mindfulness—namely, obtaining a non-judgmental attitude—in order not to exclude studies that did not address this more meta-cognitive component of the mindfulness construct cf. [38,39]. Studies were excluded if they only examined the effect of exposure to virtual or indoor nature.

#### 2.1.3. Comparators

Study groups were compared with (1) interventions without exposure to nature but with guided mindfulness, or (2) non-active control conditions (e.g., waiting lists or written materials). Studies were also included if they employed other or no comparators, in which case they were then evaluated as open trials.

#### 2.1.4. Outcomes

These were any psychological (e.g., depression), physiological (e.g., heart rate), or interpersonal (e.g., prosocial orientation) outcomes based on client-level data for which an effect size could be calculated.

#### 2.1.5. Study Design

Both open and controlled studies were evaluated. Only quantitative peer-reviewed studies reported in the English language were considered for inclusion.

### 2.2. Search Strategy

Search terms for mindfulness and nature were found in the target research articles. An abstract-based search was then conducted in electronic databases covering the natural environment and psychology. The databases searched were PsycINFO, Scopus, Web of Science, and Ovid, covering Agricola 1970 to August 2018; Ovid MEDLINE(R) Epub ahead of print, in-process, and other non-indexed citations; Ovid MEDLINE(R) Daily; Ovid MEDLINE(R) 1946 to present; Cab Abstracts 1910–2018; and Embase 1974 to present. Search terms related to mindfulness (Meditati* or mindfulness or MBSR or MBCT) were combined with search terms related to therapies used in natural settings—or “forest bathing”, Ecotherapy, “Eco therapy”, “Eco-therapy”, “Nature-based therapy”, “Nature-based therapy”, “Wilderness therapy”, “Horticultural therapy”, “Nature therapy”, “Nature involvement”, “nature-based intervention*”, “cognitive behavior therapy*”, or “Nature-assisted therapy”—and with search terms related to the natural environment and “Restorative nature”, “nature contact”, “nature exposure”, “nature-based activities”, “Restorative garden”, “Healing nature”, “Healing garden”, “Therapeutic nature”, “Therapeutic garden”, “Therapy garden”, “Care garden*”, Wilderness, Forest*, Woods, Outdoor*, “Open space*”, Park, “Green space*”, greenspace*, “Natural environment”, “Marine environment”, “Ocean wealth”, or “Blue gym”.

The search terms were defined and the searches were conducted for the period from the earliest dates available in the databases through August 2018. Additionally, a backward search (snowballing) and a forward search (citation-tracking) were conducted for the included articles. Abstracts and full texts were evaluated, and disagreements over the inclusion/exclusion of a study were resolved by consensus.

### 2.3. Data Extraction

Means and standard deviations for all quantitative outcomes were extracted. When these were not available, other test parameters were used (e.g., *F* and *p*). In cases where an effect size could not be calculated, the study’s authors were contacted. Studies were coded for participant characteristics (i.e., age, gender, and race/ethnicity), duration of intervention (from first to last intervention session), time to follow-up, and number of hours with mindfulness practice. All health outcomes were categorized as measures of physical, psychological, or interpersonal outcomes. The characteristics of nature were coded as either mixed outdoor environment containing natural elements (often with predominant built structures), garden/park with settings composed of natural elements intended to make it “appear natural”, or forest/wild nature with predominantly natural elements unaffected by human interventions [17]. The types of mindfulness were coded as formal mindfulness i.e., guided meditation, or informal mindfulness where attention is guided to the present moment during everyday activities. Formal mindfulness was coded as open-awareness meditation versus concentration meditation. Mindfulness was also coded as the intention to induce state mindfulness or build trait mindfulness [6,7]. All the coding was verified by the co-author. Data extraction protocols are available upon request from the corresponding author.

### 2.4. Quality Assessment

Studies were evaluated for quality using the quality assessment tool for quantitative studies from the Effective Public Health Practice Project (EPHPP) [40]. For each of the six components: Selection bias, study design, confounders, blinding, data collection methods, and withdrawals and dropouts, the studies were rated as strong, moderate, or weak, following the guidelines from the EPHPP tool. These ratings were, also in accordance with the EPHPP guidelines summed to create a global quality score (see Table 1). For studies with no weak ratings for the six components, the global quality score was set to be 1 = strong. Studies with one weak rating the summed score was set to be 2 = moderate. If two or more of the six components had weak scores, the global quality rating was set to be 3 = weak. Ratings were made and disagreements were discussed and resolved by consensus.

### 2.5. Analytical Overview

Meta-analyses were performed for the designs and outcomes combined, as well as separately for each of the design types and each of the outcome categories. Analyses were conducted for two time periods: From pre- to immediately post-intervention, and from pre-intervention to the last follow-up assessment. All analyses were based on random-effects models.

The planned exploratory moderation analyses of categorical variables (e.g., type of nature and type of mindfulness) were explored with meta-ANOVAs. Analyses were performed when a sufficient number of studies (k ≥ 3) was identified for a given moderator category. Continuous moderator variables (i.e., age, % women, % Whites, number of sessions with mindfulness, duration of treatment, and time to last follow-up) were considered in meta-regression analyses, based on random-effects models and estimated with the maximum likelihood method. In the regression analyses, the proposed moderators served as independent variables, with effect size serving as the dependent variable.

Effect sizes were expressed as Hedges’ *g* in order to adjust for potential bias to overestimate the effect size in small samples [41], with values of 0.2, 0.5, and 0.8 considered small, medium, and large respectively [42]. A *p*-value below 0.05 was considered statistically significant. Positive effect sizes indicate a positive effect of the interventions. Each effect size was weighted by its precision (inverse variance), so that interventions with larger samples contributed more to the estimate of the overall effect size. Heterogeneity was explored using Q and I^2^ statistics. Q-tests concern the probability that results reflect systematic between-study differences. Due to the generally low statistical power of heterogeneity tests, a *p*-value of 0.10 was used to indicate heterogeneity [43]. The I^2^ statistic is an estimate of the degree of heterogeneity, and is considered to be unaffected by the number of studies. An I^2^ value of 0% indicates no observed heterogeneity. Values of 25%, 50%, and 75% are considered low, moderate, and high respectively [44].

Positive and negative findings may not be equally likely to get published, introducing the risk of publication bias. The distribution of effect sizes was therefore visually inspected by means of funnel plots and tested with Egger’s test [45]. When the analyses indicated possible publication bias, an adjusted effect size was estimated using Duval and Tweedie’s [46] trim-and-fill method, which imputes missing results and recalculates the effect size. In addition, the file drawer problem—the possibility that unidentified or unpublished studies with null findings could alter statistically significant meta-analysis results—was evaluated by Rosenthal’s fail-safe number [47]. If the fail-safe number exceeded 5k + 10, with k being the number of studies included in the meta-analysis, the file drawer problem was considered sufficiently low to allow acceptance of the results as unaffected by that potential source of bias.

All analyses were conducted using the Comprehensive Meta-Analysis program, version 3.3.070 Eaglewood, NJ, USA: Biostat (“Comprehensive Meta-Analysis,” 2014).

## 3. Results

### 3.1. Search Results

A total of 987 publications were identified: 949 through searches in databases, and 38 from other sources. After the screening of abstracts, 841 records were excluded, resulting in 146 full-text articles; of these, 120 were excluded, primarily due to a lack of relevant content on interventions (see Figure 1). Five authors were contacted for the data necessary to calculate an effect size. Three authors provided the data. In the fourth case, the effect size was set to zero, and the study was included. The last study was excluded due to a lack of information concerning the relevant outcomes. In total, 26 articles from 25 independent studies were included. Seven interventions with mindfulness in a natural setting were compared with a similar intervention without exposure to nature (i.e., active control). Seven studies compared interventions with groups in non-active control conditions (i.e., passive control), and 13 studies were evaluated as open trials.

### 3.2. Characteristics of Studies

The characteristics of the 25 included studies are summarized in Table 1. Most studies were from Western countries (12 North American and five European), while eight were from Southeast Asia. Included in the meta-analysis were a total of 2990 participants across the studies (mean *N* per study: 120, range 19–659). The mean age was 30.71 (range 12–89), with 51.8% female (range 0–100) and 66.2% White/Caucasian participants. In five studies, physical illnesses were targeted (i.e., cancer, hypertension, coronary diseases, chronic diseases, or pain). In six studies the treatment target was a psychological diagnosis (e.g., stress, anxiety, and depression), and five studies investigated treatment for substance abuse. Eight included participants with no diagnoses. Twelve studies reported effects on psychological measures (e.g., anxiety and depression). Nine studies had physical outcome measures (e.g., heart rate variability and cortisol level), and eight studies reported effects on interpersonal measures (e.g., work function).

The EPHPP assessment tool was used to evaluate the studies’ quality [40]. Global ratings were strong for one study, fair for six, and weak for 18 studies. Most studies obtained a strong score for data collection methods (k = 22) and for reporting withdrawals and dropouts (k = 14); on the other hand, most studies obtained a weak score for selection bias (k = 13) and blinding (k = 21).

**Table 1 ijerph-16-03202-t001:** Characteristics of included studies.

Study Author, Year, Country	Population *N* = Included (Completed/Follow-up), Age = Range (Mean), Gender, Ethnicity, Target Group	Intervention Duration, Content	Comparators Duration, Content	Outcomes Measurements, Tools	Time Intervention, Timings of Data Collection	Setting Location of Study Group	EPHPP Quality
Comparator: interventions incl. mindfulness, but without nature contact							
Ballew & Omoto (2018) [48] USA	*N* = 100 Age = 18–24 (19.3) % female = 55 % Whites = 21 Study group: Students, no diagnosis	15 min in natural settings. Instructions to look at surrounding features and pay attention to all details, colors, and textures, to use all senses to take everything in. Note-taking.	Same program as study group, human-built outdoor environment.	Absorption, awe, happiness, joy, contentment (rating sentences)	T1: Survey just after 15-min. intervention	Arboretum, sitting on a bench, view of trees, creek, bamboo, etc.	3
Shin et al. (2012) [49] Korea	*N* = 69 (68) Age = 20.4 (±1.5) % female = 100 % Whites = N/A Study group: Students, no diagnosis	35 min. walk, 10 min. rest, 35 min. walk, 10 min. rest. Mindful walking with focus on breath and sensations.	Same program as study group, indoor setting.	Anxiety (STAI), self-esteem (RSE), happiness (HI-K)	T1: Before intervention T2: Just after 90-min. intervention	Undisturbed rocky area with old-growth broad-leaved evergreen trees	3
Passmore & Holder [50] (2017) Canada	*N* = 395 (364) Age = 20.09 % female = 67 % Whites = N/A Study group: Students, no diagnosis	2 weeks. Instructions to be mindful of emotions evoked by natural objects/scenes in everyday life; to describe strong emotions and take pictures of the nature that evoked them.	Same intervention as study group, human built environment. Passive controls: Continue regular routines.	Affect (PANAS), elevation (EES), meaning (SMS), connectedness (GSC), prosocial orientation (PSO), connectedness to nature (CNS), engaging with beauty (EWB)	T1: Just before intervention (PANAS only) T2: Just after 2-week intervention (all measurements)	Everyday environment of university students	3
Kim et al. (2009) [51] Korea	*N* = 73 (63) Age = 46.2 % female = 85.7 % Whites = *N*/A Study group: Major depressive disorder	4 weeks, 3 hours weekly, for CBT, positive psychology tools, and mindfulness meditation on breath, wind, forest, and sounds (insight meditation).	Same program as study group, indoor setting. Meditation focus on breath and indoor or window objects. Passive control: TAU.	Depression (BDI, HRSD, MADRS), quality of life (SF-36), stress (HRV, cortisol)	T1: Just before treatment, all measurements T2: T1 + 1 week, depression questionnaires T3: T1 + 2 weeks, depression questionnaires T4: After 3 weeks of treatment, all measurements	44-ha arboretum, 2035 species	2
Willert et al. (2014) [52] Denmark	*N* = 93 (66/49) Age = 25–59 (45.0) % female = 82.8 % Whites = N/A Study group: Stressed students	16 weeks, 5 days a week, 9 a.m. to afternoon. Groups of max. 12. All-day exercises from meditation training programs (MBCT and Five Tibetans), horticultural activities, nature walks, stress management, job counseling, individual psychotherapy sessions.	Same program as study group, indoor setting.	Stress (PSS-10), sleep (BNSQ), mindfulness (FFMQ - 3 dimensions), self-efficacy (COPSOQ-II), Outcome Rating Scale, work ability (WAI)	T1: Just before treatment T2: Just after 3 months of treatment T3: T2 + 3 months	Garden incl. greenhouse, near forest and beach	2
Vujcic et al. (2017) [53] Serbia	*N* = 30 Age = 25–65 (45.35) % female = 70 % Whites = N/A Study group: Depressed, anxious	4 weeks, 3 one-hour sessions per week of horticultural therapy, art therapy, and relaxation/meditation sessions. All main activities relate to work with living plants.	Parallel indoor activities with study group, incl. occupational, art, and conventional therapies.	Depression and anxiety (DASS21)	T1: Just before intervention T2: Just after 4 weeks’ intervention	Botanical garden incl. open space, greenhouse, Japanese garden, fountain	3
Lymeus (2018) [54] Sweden	Study 1: *N* = 89 Age = (23) % female = 64 % Whites = N/A Study 2: *N* = 51 Age = (23) % female = 72.5 % Whites = N/A Study groups: Stressed	5 weeks, 1 weekly 90-min. session. Manual-based meditation training program (REST), each session beginning and ending with 15–20 min. guided open monitoring meditation (no specific tradition). Exercises and themes. Homework: 15 min. meditation most days.	Classroom setting. Same schedule as study group. Beginning and end of sessions: Focused attention meditation (no specific tradition), exercises and themes.	Attention (LDST; TMT study 2), affect (SCAS)	Study 1: T1: Before/after session 1 T2: Before/after session 3 T3: Before/after session 5 Study 2: T1: Enrollment T2: Before/after initial 20-min. meditation in session 1 T3: Before/after initial 20-min. meditation in session 3 T4: Before/after initial 20-min. meditation in session 5	Botanical garden incl. tropical greenhouse, water bodies, orangery	3
Comparator: Non-active control conditions (see also Kim 2009 and Passmore 2017 above)							
Han (2016) [55] Korea	*N* = 61 Age = 25–49 (39.75) % female = 57.4 % Whites = N/A Study group: Chronic pain	24-hour residential intervention (noon to noon). In forest: Walking, therapeutic activities, physical exercises, mindfulness meditation. Indoor music therapy, psycho-education on stress and pain.	Usual weekend routines, except visiting natural environment or heavy loads of work.	Stress (HRV, HR) natural killer cells (NK), pain (VAS), depression (BDI), health-related quality of life (EQ-VAS)	T1: Just before treatment T2: T1 + 1 day just after 24 hours’ treatment	Foot of a mountain: forest valley, “spectacular” views	3
Won (2012) [56] Korea	*N* = 92 Age = 45.26 % female = 8.7 % Whites = N/A Study group: Detoxified chronic alcoholics	9 days: 3 days actively interacting with nature, 3 days challenging activities in nature, 3 days activities for introspection (nature meditation, counseling in nature etc.).	Inpatients, no specific treatment described.	Depression (BDI)	T1: Just before treatment T2: T1 + 9 days just after treatment	2140-ha recreational forest	2
Warber et al. (2011) [57] USA	*N* = 58 (47/41) Age = 25–75 (59.3) % female = 40.4 % Whites = 85.1 Study group: Coronary syndrome	4-day residential program. Study group 1 (MFTE): Meditation, guided imagination, journaling, drawing, nature activities, nature imagination. Study group 2 (LCP): Nutrition, physical exercise, stress management based on mindfulness and whole-person approach. Both groups: Telephone coaching biweekly for 3 months.	No treatment.	Depression (BDI, BSI), stress (PSS), hope (SHS), gratitude (SG), quality of life (SF-36), spirituality (ISWBS), personal transformation (TCQ), physical activity (PPAQ), stress (HR, BP, BMI, lipid levels, lipid particle size, high sensitivity C-reactive protein, biomarkers IL-6 and IL-10)	T1: Just before treatment T2: Just after 4 days’ treatment T3: T2 + 3 months T4: T2 + 6 months Biophysical measurements only at T1 and T2	“Beautiful” rural settings	3
Sung et al. (2012) [58] Korea	*N* = 56 Age = 66.0 % female = 60.7 % Whites = N/A Study group: Hypertension stage 1	3-day forest therapy program: Relationship-building, stress and health management, mindfulness and gratitude meditation in forest.	Written material on hypertension management.	Stress (BP), salivary cortisol level, quality of life (QoL, 5 dimensions)	T1: Just before intervention, all measurements T2: Just after 3-day intervention, all measurements Follow-up: Self- monitoring BP, 8 weeks	Recreational forest in mountain region	2
Passmore (2014) [59] Canada	*N* = 86 (84) Age = 18–45 (20.96) % female = 86.9 % Whites = N/A Study group: Students, no diagnosis	14 days. Written instructions to immerse themselves in nature activities whenever possible in everyday lives. Keeping logbook of nature activities for each day.	Solving anagram puzzles whenever possible in their everyday lives.	Affect (PANAS), elevation (EES), meaning (SMS), motivation (SCM)	T1: Just before intervention T2: T1 + 14 days just after intervention	Everyday environment of Canadian students	1
Studies with no comparators relevant for this review							
Jung (2015) [60] Korea	*N* = 19 Age = 29.4 % female = 100 % Whites = N/A Study group: No diagnosis	2 days, noon day 1 to noon day 3. Indoors: lectures on coping with stress, counseling, cognitive therapy, music therapy. In forest: 5 hours’ meditation, walking, exercises.		Stress (HR, HRV, cortisol), natural killer cell activity (NK), burnout (MBI-GS), stress (WSRI), recovery (REQ)	T1: Just before intervention T2: T1 + 2 days just after intervention T3: T2 + 14 days All measures at T1 and T2, except MBI-GS second measure at T3	2140-ha recreational forest	3
Yu et al. (2017) [61] Taiwan	*N* = 128 Age = 45–86 (60) % female = 65.6 % Whites = N/A Study group: 46% chronic diseases (e.g., diabetes)	2 hours, 2.5 km sensory forest walk incl. guided stimulation of senses (visual, auditory, olfactory, tactile). Groups of 6–12 participants.		Mood (POMS-SF), anxiety (STAI), stress (pulse rate, BP, HR, HRV)	T1: Just before intervention T2: Just after 2-hour intervention	Sensory forest, in valley surrounded by mountains on three sides (part of Xitou Nature Education Area in Japan)	3
Korpela et al. (2017) [62] Finland, Luxembourg, Sweden	*N* = 283 Age = 13–82 (47.2) % female = 74 % Whites = N/A Study group: No diagnosis	Well-being trails in the 3 countries, 4.4–6.6 km, containing the same 9 signposts with tasks: Self-monitoring (first and last), relaxation, letting oneself be fascinated, observing nature and one’s own body and mood.		Restorative change (4 items), mood (1 item), nature connectedness (3 items)	T1: At first signpost on the trail T2: At last signpost on the trail	Hiking tracks incl. forests, lakesides, fields, cultural landscapes	3
Yang (2018) [63] USA	*N* = 29 (27/26) Age = 66–89 (73.2) % female = 83 % Whites = 79 Study group: No diagnosis	4 weeks, 8 sessions of 30 min. individual mindful walking. Before walking, guidance either to become familiar with the environment, to focus on breath or movement, or to scan through the body.		Affect (PROMIS), mindfulness (SMS)	T1: Just before treatment T2: Just after 30 min. mindful walking	Flat designated route in arboretum	2
Corazon et al. (2018) [64] & Stigsdotter (2018) [65] Denmark	*N* = 43 (42/29) Age = 47.9 % female = 81.6 % Whites = N/A Study group: Severely stressed	10-week nature-based therapy, 3 times, 3 hours per week. Activities individually and in groups: Exercises in accordance with MBSR and related to nature experiences, such as mindful walking in natural setting. Gardening and relaxation/reflection time. Individual therapeutic sessions (CBT) and support for return to work.		Sick leave and contact with GP (from national database), well-being (PGWBI), burnout (SMBQ)	T0-T1: 1-year time span T1: Treatment start T2: Just after 10 weeks’ treatment T3: T2 + 3 months T4: T2 + 6 months T5: T2 + 12 months	1.4-ha wild forest garden located in larger arboretum	2
Sahlin et al. (2014) [66] Sweden	*N* = 44 (33) Age = N/A % female = 100 % Whites = N/A Study group: Stressed	12 weeks, 3 hours weekly. 3 intervention groups Activities: Walks, relaxation, mindfulness, therapeutic painting, group therapy, information about stress and health, garden and nature activities.		Burnout (SMBQ), work ability I (WAI, adjusted), stress (scale tools created for this study), sleep (KSQ)	T1: Just after first session T2: Just after 12-week program T3: T2 + 6 months T4: T2 + 12 months	225-ha wild nature, incl. forest, ponds, moorland, hills; wooden house, greenhouse	3
Nacau et al. (2013) [67] Japan	*N* = 22 Age = 58.2 % female = 81.8 % Whites = N/A Study group: Cancer, after treatment	12 weeks, once per week, 6 hours. 40 min. walks, 60 min. horticultural therapy, 90 min. indoor yoga and meditation (1 session), 60 min. supportive group therapy (5 sessions). Homework: yoga (video).		Well-being (FACIT) incl. physical, cancer fatigue (CFS), quality of life (SF-36), mood (POMS-SF), anxiety (STAI), natural killer cell activity (NK)	T1: Just before treatment T2: Just after 12 weeks’ treatment	Park incl. forest, streams, lawns, gardens; yoga and meditation indoors in the park	3
Combs et al. (2016) [68] USA	*N* = 704 (659) Age = 16 % female = 32 % Whites = 85 Study group: behavior/ substance/ mood issues	90-day program. Nomadic hiking and/or expeditions and tasks associated with outdoor living. Therapeutic tools: The wilderness itself, CBT, choice therapy, family systems, mindfulness techniques, diet, physical exercise. Individual/ group therapy sessions twice a week.		Psychological and behavioral symptoms and social functioning (Y-OQ_SR)	T1: At intake T2: T1 + 3 weeks T3: T1 + 5 weeks T4: At discharge T5: T4 + 6 months T6: T4 + 18 months.	Wilderness in undeveloped areas	3
Russell (2016) [69] Canada	*N* = 43 (32) Age = 18–24 (22.9) % female = 0 % Whites = N/A Study group: Substance abuse	90-day, 10-bed outdoor behavioral healthcare program (Shunda Creek), incl. weekly 1–5-day adventure trips integrating mindfulness-based experience (MBE) with psychotherapy.		Subjective discomfort, interpersonal relations, social roles (OQ-45.2), mindfulness (FFMQ)	T1: At intake T2: At discharge (average T1 + 93.7 days)	Wild nature, incl. mountains	3
Russell et al. (2017) [70] USA	*N* = 168 Age = 21.5 % female = 0 % Whites = 40 Study group: Substance abuse	90-day outdoor behavioral healthcare program (Shunda Creek): Weekly 1–5-day adventure trips integrating MBE with psychotherapy.		Helpfulness and mindfulness (subscales of OQ-45.2), adventure therapy experience (ATES)	T1: At admission T2–T13: Every second week until discharge	Wild nature, incl. mountains	3
Russell (2018) [71] Canada	*N* = 57 (46) Age = 12–17 (16.6) % female = 43.9 % Whites = 57.9 Study group: 74% diagnoses, ADHD/substance use	8-week, 15-bed program: family therapy, daily individual/group therapy, educational programming. Base camp model: Adventure therapy and development of mindfulness skills.		Emotional and behavioral symptoms (Y-OQ SR 2.0), mindfulness (CAMM)	T1: Just before treatment T2: Just after 8 weeks’ treatment	Wild nature, incl. mountains	3
Chapman et al. (2018) [72] Canada	*N* = 177 Age = 18–24 (21.5) % female = 0 % Whites = 42.1 Study group: Substance use	90-day outdoor behavioral healthcare program (Shunda Creek): Weekly 1–5-day adventure trips integrating MBE with psychotherapy.		Subjective discomfort, interpersonal relations, social roles (OQ-45.2)	T1: At intake T2: At discharge (average T1 + 79.6 days)	Wild nature, incl. mountains	3
Unsworth et al. (2016) [73] study 2 USA	*N* = 39 Age = 21 % female = 64.1 % Whites = N/A Study group: No diagnosis	3 days’ Aztec adventure camp in nature, incl. 15 min. formal daily morning meditation, and encouragement to continue mindfulness practice throughout the day.		Self-nature interconnectedness, nature in self (INS), mindfulness (FMI)	T1: Just before treatment T2: Just after 3 days’ treatment	Wild nature	3

Outcome measures: Beck Depression Inventory (BDI), Hamilton Rating Scale for Depression (HRSD), Montgomery-Åsberg Depression Rating Scale (MADRS), Depression Anxiety Stress Scale (DASS21), Workers Stress Response Inventory (WSRI), State-Trait Anxiety Inventory (STAI), Perceived Stress Scale (PSS), Swedish Core Affect Scale (SCAS), Positive and Negative Affect Scale (PANAS), Psychological General Well-Being Index (PGWBI), Patient Reported Outcomes Measurement Information System (PROMIS), Profile of Mood States short form (POMS-SF), Rosenberg’s Self-Esteem Scale (RSE), self-efficacy scale from Copenhagen Psychosocial Questionnaire (COPSOQ-II), Recovery Experience Questionnaire (REQ), Happiness Index for Koreans (HI-K), Elevation Experience Scale (EES), Sense of Meaning Scale (SMS), State Hope Scale (SHS), Gratitude Scale (GS), Self-Concordant Motivation (SCM), General Sense of Connectedness (GSC), Five-Facet Mindfulness Questionnaire (FFMQ), Child and Adolescent Mindfulness Measure (CAMM), Freiburg Mindfulness Inventory (FMI), State Mindfulness Scale (SMS), Cancer Fatigue Scale (CS), Shirom-Melamed Burnout Questionnaire (SMBQ), Malash Burnout Inventory-General Survey (MBI-GS), Visual Analog Scale (VAS), Brief Symptom Inventory (BSI), Functional Assessment of Chronic Illness Therapy-Spiritual Well-Being Scale (FACIT), Irvine’s Spiritual Well-Being Scale (ISWBS), Transmutation Change Questionnaire (TCQ), Short Form-36 to measure health-related quality of life (SF-36), Quality of Life (QoL), EuroQol Visual Analog Scale (EQ-VAS), Basic Nordic Sleep Questionnaire (BNSQ), Karolinska Sleep Questionnaire (KSQ),Work Ability Index (WAI), Paffenbarger Physical Activity Questionnaire (PPAQ), Prosocial Orientation (PSO), Connectedness to Nature Scale (CNS), Engaging with Beauty Scale (EWB), Inclusion of Nature in Self (INS), Outcome Rating Scale (everyday functioning), Outcome Rating Scale Youth Outcome Questionnaire Self-Report (Y-OQ_SR), Outcome Questionnaire measuring psychological and behavioral symptoms (OQ-45.2), Adventure Therapy Experience (ATES), Letter-Digit Substitution Test (LDST), Trail-Making Test (TMT), heart rate (HR), heart rate variability (HRV), body mass index (BMI), lipid levels, lipid particle size, high sensitivity C-reactive protein, biomarkers IL-6 and IL-10, salivary cortisol level, pulse rate, systolic and diastolic blood pressure (BP), natural killer cell activity calcein-AM release assay using NK-sensitive K-562 cells as a target (NK).

### 3.3. Intervention Characteristics

The interventions were highly heterogeneous concerning length, setup, and content, as well as the amount and type of mindfulness, and the choice of natural setting. All studies had psychological endpoints, and the most prevalent outcomes were psychological wellbeing or positive emotions (k = 14), attention (k = 7), depression (k = 5), and anxiety (k = 4). Physiological endpoints were reported in eight studies, all including cardiovascular system outcomes, e.g., heart rate variability and blood pressure, and four studies reported outcomes related to the immune system, investigating e.g., natural killer cells and inflammation. Of studies including interpersonal outcomes (k = 7), four reported on interpersonal functioning and three on workability. The length of intervention varied from 15 minutes to 90 days, with follow-up data available for eight studies. The follow-up time ranged from two weeks to 18 months post-treatment.

Three main types of intervention emerged:Short single-instruction intervention studies (k = 7) aimed at healthy participants, who were guided either to be mindful on their own while sitting or walking, or to be more extensively mindful in their everyday lives.Weekly meetings (once or more per week) targeting stressed, anxious, or depressed people (k = 6), mostly with gardening activities and psychotherapy. One study stood out in this format as only containing meditation training.Residential interventions (k = 11), of which five were wilderness therapy of several weeks’ duration. Participants in these studies were young people, mostly males diagnosed with substance use disorders. The other six residential interventions were shorter and in diverse settings.

Among the interventions reporting on formal meditation (k = 8), three had full meditation training protocols: MBSR, MBCT [74], and restoration skill training (ReST) [54]. The MBSR and MBCT were integrated in extended nature-based interventions, with only ReST being a stand-alone program. Five studies mainly included concentration meditation, while open-awareness meditation was exclusively practiced in two programs. Among the informal mindfulness interventions (k = 8), four gave brief guidance toward the present moment, and three investigated a specific program of mindfulness-based experience [69] that integrated psychotherapy with mindfulness on adventure trips. The interventions’ natural environments varied widely, from designed small gardens to vast wild nature areas. The amount of time devoted to guided mindfulness was less than five hours in all but two studies.

### 3.4. Pooled Effect Sizes and Between-Study Differences

The overall combined effect size from pre- to post-treatment across outcomes and designs was significant and of medium size (*g* = 0.54, *p* < 0.001; see Table 2). Studies employing an open, non-controlled design, again across outcomes, revealed a significant effect of medium size (*g* = 0.66, *p* < 0.001); the same was true for studies with passive control groups (*g* = 0.58, *p* < 0.001). Studies using active control groups also revealed a significant effect, albeit small in size (*g* = 0.26, *p* = 0.023). Only one intervention was found that had a natural setting but without mindfulness as an active control group, and only one intervention included manual-based stand-alone mindfulness conducted in nature. The effects of these types of intervention were therefore not calculated.

When the effects from pre- to post-treatment were evaluated for individual outcomes across designs, the effect on psychological outcomes was significant and of medium size (*g* = 0.55, *p* < 0.001). The effects on social (*g* = 0.39, *p* = 0.004) and physical (*g* = 0.36, *p* = 0.011) outcomes were significant and of small size.

Concerning effects at follow-up, the combined effect across designs and outcomes was significant and of moderate size (*g* = 0.56, *p* < 0.001). Only the studies employing an open trial design were sufficient in number to perform a separate meta-analysis, which also revealed a significant effect of medium size (*g* = 0.66, *p* < 0.001).

A number of planned moderation analyses were conducted, none of which were statistically significant. However, type of nature (*p =* 0.068, *Q* = 3.4) and trended toward significance. Interventions in wild/forest environments obtained a numerically larger effect (*g* = 0.66) than interventions in garden/park environments (*g* = 0.33). Moreover, the type of mindfulness (*p* = 0.078, *Q* = 0.31) did as well trend towards significance, and informal mindfulness interventions obtained a numerically larger effect (*g* = 0.80) than formal mindfulness interventions (*g* = 0.37). The effect size for inducing state mindfulness was larger (*g* = 0.62) than for building trait mindfulness (*g* = 0.10). No significance was found for this moderator (*p* = 0.107). Only two interventions primarily used open awareness, and the moderating effect of this type of mindfulness was therefore not calculated (see Table 3).

### 3.5. Publication Bias

For all statistically significant results, publication bias was evaluated. Eight analyses were adjusted for possible publication bias, which in only one case (i.e., effect on social outcomes) changed the result from significance to non-significance. Seven out of 13 studies failed to meet the criterion for the fail-safe *N*, indicating lack of robustness for approximately half of the results.

## 4. Discussion

We were able to identify 25 independent studies that met our criteria. Across the designs of these studies, an initial synthesis showed overall positive effects of mindfulness in natural settings evaluated in both open trials and controlled trials using non-active control groups. These results support our hypothesis that context may play a significant role in the benefits of mindfulness-based interventions. This may be explained by the experience of the natural environment, which is so fascinating that it calls for soft attention, thereby allowing disengagement [4]. This is comparable to “letting go” in mindfulness, where the meditator is guided not to mentally hold on to anything, and not to push anything away [6]. Another explanation could be that natural stimuli occupy the attention [4] and consequently reduce the tendency for the mind to wander [5], which is another aim in mindfulness training. While experienced meditators may be able to stay present during meditation, exposure to nature may support inexperienced or otherwise challenged meditators who would otherwise be at risk of losing concentration completely or becoming emotionally overwhelmed [4].

A number of moderation analyses were conducted. Forests/wild nature and informal mindfulness were found to trend-wise increase positive health outcomes based on large differences in effect size, although these were not significant. The lack of significance may be due to the low number of studies, and should be interpreted according to the effect sizes. The numerically larger positive effect of natural settings characterized as wild supports the findings by Grahn and Stigsdotter [75], suggesting that stressed individuals prefer natural environments that are wild and untouched and offer a variety of species but are still experienced as safe.

Informal mindfulness tended to moderate positive health outcomes compared with formal mindfulness, as did inducing state mindfulness compared with building trait-mindfulness. Both may be explained by the outward focus, which may be more beneficial in a natural setting, as it allows more contact with nature. Furthermore, the possibility of engaging in activities during informal mindfulness may explain the more positive outcomes. Corazon, Schilhab, and Stigsdotter [76] argue that bodily involvement with the environment is important in nature-based therapy, as it strengthens the memory of experiences in nature, and thereby prolongs and confirms the therapeutic effect.

### 4.1. Implications for Research and Practice

The field of nature-based mindfulness is in its infancy, and is not yet defined; our study only suggests some structure. One of the aspects that are still in need of investigation is whether certain types of mindfulness are more suited than others for training and use in natural settings, and whether this depends on the characteristics of the natural setting and other components of the intervention. Informal mindfulness is compatible with ordinary activities [5], such as walking a forest trail e.g., [62], and has been shown to be a tool for healthy people to enhance the positive effects of contact with nature [63,77]. Forest bathing is a research field that addresses this [25], but a systematic approach to mindfulness is still needed in this context. In nature-based therapy, on the other hand, informal mindfulness enhances awareness of negative thought patterns, which seem easier to detect in natural settings, and this is of value in a therapeutic context [66,78,79].

Lymeus et al. [54] argue that the practice of open-monitoring meditation (comparable to open-awareness meditation) in natural environments is superior to concentration meditation, as it allows natural stimuli softly and effortlessly to hold the attention to the present moment. Due to the scarcity of available studies, it was unfortunately not possible to compare open-awareness meditation with concentration meditation as moderators of positive health outcomes in this review. It is recommended that future studies should address this gap in knowledge, and should also carefully define and describe the way the mindfulness is conducted and the characteristics of the natural setting in which the therapy takes place. In addition, it seems reasonable to not only include nature in health promoting activities [23], but also to include informal mindfulness (i.e., guided attention to the senses with an attitude of non-judgment and openness) in nature-based therapy. Formal meditation in natural settings also seems to be a promising tool, and further research is needed to provide guidelines for such practice.

### 4.2. Limitations

The rather low quality of the included studies poses a threat to the validity of the findings, which need to be confirmed in high-quality studies. In particular, blinding and selection bias were issues, and only a few trials could be categorized as clinical trials according to the EPHPP assessment criteria. With only 25 studies included in this review, and in light of the heterogeneity of the participants and intervention characteristics, the generalizability is limited. Furthermore, a different definition of mindfulness than that employed in this study might affect the character and number of studies included. The included studies would preferably define mindfulness as containing an attitude of e.g., warmth and non-judgment, but meditation practices are rarely described in detail, and such narrow inclusion criteria would at present exclude most studies.

## 5. Conclusions

This systematic review and meta-analysis shows that nature-based mindfulness has had a positive effect on psychological, physical, and social conditions. Furthermore, nature-based mindfulness is moderately superior to mindfulness conducted in non-natural settings. However, at this point we know very little about the effect of different types of mindfulness, and more research is needed to understand what an optimal mindfulness intervention in a nature-based setting should consist of. Mindfulness in wild nature seems to be more beneficial than mindfulness in more cultivated settings, but the importance of the setting needs further investigation.

## Figures and Tables

**Figure 1 ijerph-16-03202-f001:**
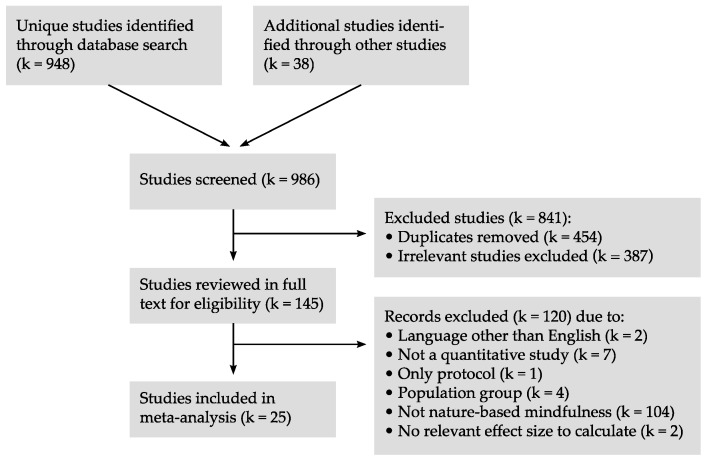
Flow chart of literature search.

**Table 2 ijerph-16-03202-t002:** Results from overall and subgroup meta-analyses.

	Sample Size		Heterogeneity				Global Effect Aizes			Fail-Safe N ^c^	Criterion
	k	*N*	Q ^b^	df	*p*	I^2^	Hedges’ *g*	95% CI	*p*		
**Outcome from pre- to post-treatment**											
Combined effect across designs and outcomes	25		241.1	24	<0.001	90.0	0.54	0.34–0.75	<0.001	2146	135
*Adjusted for publication bias ^a^*	32						0.83	0.55–0.91			
Open trials	13	1737	170.4	12	<0.001	93.0	0.66	0.38–0.94	<0.001	1211	75
*Adjusted for publication bias*	15						0.76	0.50–1.02			
Studies with passive control group	6	821	10.3	5	0.068	51.4	0.58	0.34–0.82	<0.001	67	45
Studies with active control group	7	900	12.1	7	0.096	42.3	0.26	0.04–0.49	0.023	11	50
**Combined effect across designs for each type of outcome**											
Psychological	24	2990	215.7	23	<0.001	89.3	0.55	0.36–0.74	<0.001	2169	130
*Adjusted for publication bias*	30						0.69	0.52–0.87			
Physical	7	439	18.6	6	0.005	67.7	0.36	0.08–0.63	0.011	29	45
*Adjusted for publication bias*	1						0.29	0.02–0.55			
Social	4	432	3.9	2	0.143	48.6	0.39	0.13–0.65	0.004	13	25
*Adjusted for publication bias*	5						0.22	−0.03–0.48			
**Outcome from pre-treatment to follow-up**											
Combined effect across designs	8	1071	13.5	7	0.060	48.3	0.56	0.34–0.78	<0.001	97	50
*Adjusted for publication bias*	11						0.73	0.59–0.86			
Open trials	4	791	8.9	4	0.064	55.2	0.66	0.39–0.92	<0.001	66	35
**Categorical moderators**											
Type of nature:											
Garden/park	8	501	17.3	8	0.027	53.7	0.33	0.09–0.56	0.008	28	55
Wild/forest	15	1578	169.5	13	<0.001	92.3	0.66	0.40–0.93	<0.001	14	80
*Adjusted for publication bias*	17						0.80	0.56–1.04			
Between-group difference			3.4	1	0.065						
Type of mindfulness:											
Formal	9	544	18.1	8	0.021	55.7	0.37	0.15–0.59	0.001	53	55
*Adjusted for publication bias*	10						0.31	0.08–0.53			
Informal	8	1309	127.6	7	<0.001	94.5	0.80	0.38–1.23	<0.001	463	50
Between-group difference			3.1	1	0.078						
Trait building	4	276	14.8	3	0.002	79.8	10	−0.49–0.69	0.732		
State inducing	21	2624	214.6	20	<0.001	90.7	0.62	0.41–0.83	<0.001	2041	115
Between-group difference			2.6	1	0.107						

^a^ The possibility of publication bias was examined with funnel plots and Egger’s tests followed by imputation of missing studies. (k) = k + number of imputed studies. ^b^ For the Q-statistic, *p*-values of <0.05 are considered indicative of heterogeneity. ^c^ The fail-safe *N* was calculated for statistically significant findings to examine the robustness of these findings, representing the number of non-significant studies that would bring the *p*-value to non-significance (i.e., *p* > 0.05).

**Table 3 ijerph-16-03202-t003:** Results from meta-regression-based moderation analyses.

Moderator	B	SE	*p*
*Participant characteristics*			
Mean sample age	<0.01	0.01	0.893
% women	<−0.01	<0.01	0.425
% Whites	0.01	0.01	0.506
*Intervention characteristics*			
Intervention duration	<0.01	<0.01	0.716
Sessions with mindfulness	<−0.01	0.01	0.597

Note: B = Unstandardized beta coefficient; SE = Standard error of B; *p* = level of significance.

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
