# Peer review of "A Systematic Review and Meta-Analysis of Nature-Based Mindfulness: Effects of Moving Mindfulness Training into an Outdoor Natural Setting"

_ijerph, 2019, doi:10.3390/ijerph16173202_

Round 1

Reviewer 1 Report

Thank you for the opportunity to review your work.  Please see the suggested comments.

This is well written paper and easy to follow.

Lines 144 – 192: it is helpful to describe the process, but it might not be useful to explain each author’s role and tasks. This section should not show “who” did what, but “how” team did for this study. If necessary, you could add more at “Author Contribution” section.

Line 387 -388: what is the difference between low quality and high quality?

Reviewer 2 Report

This was a excellently written paper on an important emerging topic.  This field will benefit greatly from this publication.  Very well done.

Only one recommendation: FlowChart in Figure 1 needs to be fixed.  Otherwise flawless.

Reviewer 3 Report

Thank you for opportunity for reviewing this paper “A systematic review and meta-analysis of nature-based mindfulness: Effects of moving mindfulness training into an outdoor natural setting”.  It is very interesting paper. However, some may need to be clarified so that the paper will be more useful.

 In the aims of your study, you stated that you want to explore the potential effect of type of natural setting and type of mindfulness practice. Would you please report your findings in this study purposes? In addition, it may be useful if you could report more specific details about the outcomes of nature-base mindfulness from your analysis. For instance, what psychological aspects did you find to be effective after practicing nature-based mindfulness: quality of life, reduce depression, etc.? Lastly, in the conclusion section: Please give readers more details about how to apply the findings of this study into clinical practices.

Round 2

Reviewer 3 Report

The authors have enough addressed the reviewer’s comments. So it is ready for publication.